# ON EXTRAPOLATION IN MATERIAL PROPERTY RE-GRESSION

## ABSTRACT

Deep learning methods have yielded exceptional performances in material property regression (MPR). However, most existing methods operate under the assumption that the training and test are independent and identically distributed (i.i.d.). This overlooks the importance of extrapolation - predicting material properties beyond the range of training data - which is essential for advanced material discovery, as researchers strive to identify materials with exceptional properties that exceed current capabilities. In this paper, we address this gap by introducing a comprehensive benchmark comprising seven tasks specifically designed to evaluate extrapolation in MPR. We critically evaluate existing methods including deep imbalanced regression (DIR) and regression data augmentation (DA) methods, and reveal their limitations in extrapolation tasks. To address these issues, we propose the Matching-based EXtrapolation (MEX) framework, which reframes MPR as a material-property matching problem to alleviate the inherent complexity of the direct material-to-label mapping paradigm for better extrapolation. Our experimental results show that MEX outperforms all existing methods on our benchmark and demonstrates exceptional capability in identifying promising materials, underscoring its potential for advancing material discovery.

## 1 INTRODUCTION

Material property regression (MPR), the task of predicting continuous material property values, plays a critical role in material discovery across diverse applications such as catalysts and batteries. Traditional density functional theory (DFT)-based methods, while accurate, are often computationally prohibitive for large-scale screening. To address this challenge, deep learning models (Xie & Grossman, 2018; Schütt et al., 2021; Yan et al., 2022; Liao et al., 2024; Shoghi et al., 2024) have emerged as efficient alternatives, providing rapid predictions that facilitate the identification of promising material candidates for further validation through detailed simulations or experiments.

Predicting material properties beyond the range covered by training data, known as extrapolation, is a crucial yet under-explored area in deep learning-based MPR. Materials scientists strive to discover materials with superior properties compared to existing ones, such as organic light-emitting diodes (OLEDs) with extreme color purity (Xu et al., 2020; Kim & Yasuda, 2022), semiconductor materials with extraordinary thermodynamic stability (Castelli et al., 2012a;b) (Figure 1), and more. In this context, the extrapolation ability of deep learning models becomes crucial, as these novel properties often do not exist in currently known materials. However, most existing MPR benchmarks (Dunn et al., 2020; Choudhary et al., 2024; Chang et al., 2022) assume that both training and testing set are independent samples from an identical distribution (i.e. *i.i.d.* samples), limiting exploration of extrapolation challenges.

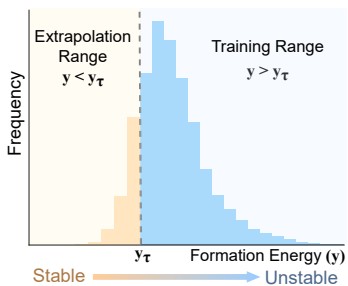

Figure 1: Extrapolation in MPR (for Formation Energy), which aims to generalize to label values ($y < y_\tau$) outside the training label range ($y > y_\tau$).

To address this gap, in this paper, we curate a comprehensive extrapolation benchmark consisting of seven datasets sourced from `Matminer` (Ward et al., 2018), accompanied by well-defined train/validation/test splits based on real-

world material applications. We then carefully evaluate the existing methods under a wide range of (1) backbones, including representative equivariant geometric GNNs such as PaiNN (Schütt et al., 2021) and EquiformerV2 (Liao et al., 2024); (2) training algorithms, including classic ERM, deep imbalanced regression (DIR) methods (Yang et al., 2021; Gong et al., 2022; Ren et al., 2022; Keramati et al., 2024), and data augmentation techniques (Yao et al., 2022; Kaufman & Azencot, 2024); and (3) metrics, such as mean absolute error (MAE) and error geometric mean (GM). Given that structure-to-property mappings are governed by intricate quantum mechanical interactions, and neural networks often struggle to capture complex non-linearity beyond the scope of training data (Xu et al., 2021), it is unsurprising that these methods struggle with extrapolation tasks in MPR, highlighting the need for more tailored methodologies for this challenge.

In response, we propose **M**atching-based **EX**trapolation (MEX), a novel framework that reframes MPR as a material-property matching problem, aimed at simplifying the complexity of target functions to enhance model extrapolation. Our intuition is that matching - focusing on the proximity between material and property representations rather than precise value predictions - reduces learning difficulty and improves extrapolation. Specifically, MEX employs two complementary training objectives to learn aligned feature spaces for material and property representation matching. First, it performs absolute matching optimization using negative cosine similarity loss, which pulls paired material and label representations closer together. Second, MEX leverages Noise Contrastive Estimation (NCE) (Gutmann & Hyvärinen, 2010) to force the model to distinguish between target and noisy labels, thus capturing fine-grained relative matching relationships. Within the well-aligned latent spaces, MEX predicts by optimizing for the nearest target value for a given sample. Experiments show that MEX not only achieves the best performance on our benchmark but also exhibits extraordinary detection capability for promising materials, demonstrating superior extrapolation capabilities and potential for more robust material discovery.

Our contributions are summarized as follows:

- We highlight the critical importance of extrapolation in MPR, an area that has been previously understudied yet holds significant implications for realistic material design scenarios.

- We curate a comprehensive benchmark specifically designed to evaluate extrapolation in material properties regression, and thoroughly investigate the effectiveness of deep imbalanced regression (DIR) and regression data augmentation (DA) methods on extrapolation tasks, revealing their limitations in handling the complexities of MPR.

- We propose MEX, a simple yet effective framework that substantially enhances extrapolation capabilities, achieving state-of-the-art performance on our benchmark.

## 2 RELATED WORK

**Material property prediction.** Recent years have witnessed the tremendous impact of deep learning on predicting material properties (Schütt et al., 2018; Yan et al., 2022; Shoghi et al., 2024). Considering the 3D atomic systems' essence of material data, numerous studies have aimed to enhance neural architectures to effectively capture the intrinsic physical symmetries of such data. SchNet (Schütt et al., 2018) and CGCNN (Xie & Grossman, 2018) pioneered the use of graph neural networks for 3D atomic systems, which modeled the pairwise atomic distance variant with regard to Euclidean transformations. Since then, a body of research has focused on encoding higher-order geometric invariants (Klicpera et al., 2020; Gasteiger et al., 2021; Yan et al., 2022) and equivariants (Schütt et al., 2021; Passaro & Zitnick, 2023; Liao et al., 2024).

Another area of focus lies in pre-training to learn transferable material representations (Zhang et al., 2023; Shoghi et al., 2024; Song et al., 2024). For instance, Shoghi et al. (2024) propose joint pre-training on force and energy prediction tasks across different chemical domains and show impressive transfer performance to downstream tasks. Song et al. (2024) employed a self-supervised pre-training task via crystal structure reconstruction based on diffusion models. Orthogonal to existing research efforts, our work focuses on the overlooked issue of extrapolation in MRP and approaches it from a unique training strategy perspective, which can use any model architecture and pre-trained model as backbones.

**Imbalanced regression.** Imbalanced regression aims to learn continuous targets from imbalanced data where certain target values are scarce, and generalize to the entire target range. Early

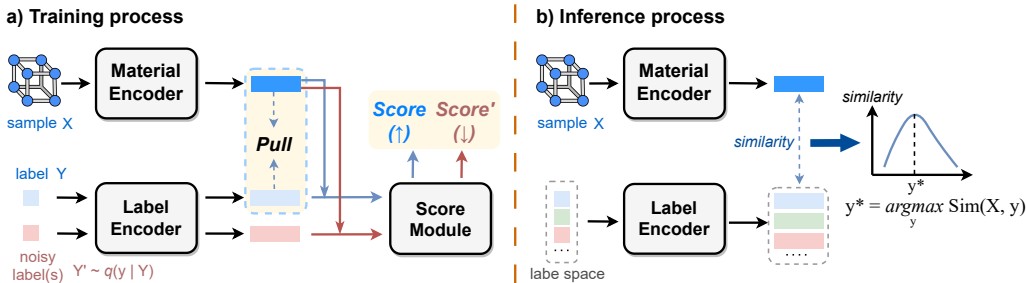

Figure 2: The framework of MEX. (a) MEX begins by drawing noisy labels from a noise distribution. Both samples and labels are embedded into the feature space, where MEX pulls the sample and its target label closer together. Noise Contrastive Estimation loss is then applied to refine this feature space by maximizing the score between the sample and its correct label while minimizing the scores between the sample and noisy labels. (b) MEX predicts the label by identifying the most similar one to the sample in the learned feature space.

works (Torgo et al., 2013; Branco et al., 2017) use over-sampling techniques by synthesizing samples for minority targets. DenseWeight (Steininger et al., 2021) and LDS (Yang et al., 2021) adopted a similar approach of using kernel density estimation to estimate the 'real' label density distribution, and subsequently re-weighting the samples accordingly. BalancedMSE (Ren et al., 2022) identifies the label imbalance that MSE carries into prediction and mitigates it by restoring a balanced prediction distribution. Current state-of-the-art approaches encourage preserving label-space relationships in the feature space, such as label similarity orders (Gong et al., 2022), relative similarities (Keramati et al., 2024) and topology (Zhang et al., 2024). Several DIR methods (Yang et al., 2021; Gong et al., 2022) have considered extrapolation as a specific DIR scenario and claimed effectiveness in this context. However, they lack dedicated research tailored to handling disjoint target label intervals, making them suboptimal for extrapolation. This work explicitly focuses on this challenge and proposes a novel training scheme for MPR by matching materials and properties within aligned feature spaces, moving beyond conventional single-point estimation employed by existing DIR methods.

## 3 METHODOLOGY

### 3.1 PROBLEM DEFINITION

We define MPR extrapolation tasks as predicting unobserved material property values that lie outside the training label range. Formally, let the input space and label space be denoted as $\mathcal{X}$ and $\mathcal{Y}$, where $\mathcal{X}$ contains the structural data of materials, and $\mathcal{Y} \subset \mathbb{R}$ corresponds to a continuous range of labels. The training domain and target domain are respectively defined as $\mathcal{D}_{\text{train}} = \{(x, y) \mid (x, y) \in \mathcal{X} \times \mathcal{Y}_{\text{train}}\}$ and $\mathcal{D}_{\text{target}} = \{(x, y) \mid (x, y) \in \mathcal{X} \times \mathcal{Y}_{\text{target}}\}$, where $\mathcal{Y}_{\text{train}}$ and $\mathcal{Y}_{\text{target}}$ are two disjoint subspaces of $\mathcal{Y}$, i.e.,

$$\mathcal{Y}_{\text{target}} \subset \{y \in \mathcal{Y} \mid y > \max(\mathcal{Y}_{\text{train}}) \vee y < \min(\mathcal{Y}_{\text{train}})\}$$

The goal of extrapolation is to learn a model $f : \mathcal{X} \to \mathcal{Y}$ that minimizes the extrapolation error $\mathbb{E}_{(x,y) \sim \mathcal{D}_{\text{target}}} [\ell(f(x), y)]$, where $\ell : \mathbb{R} \times \mathbb{R} \to \mathbb{R}$ is the loss function. Note that the model can only utilize $\mathcal{D}_{\text{train}}$ without further adapting to $\mathcal{D}_{\text{target}}$ during training.

### 3.2 MATCHING-BASED EXTRAPOLATION

In contrast to directly mapping materials to properties, we argue that learning the matching relationship between them presents a simpler learning target, facilitating model generalization in previously unseen label ranges. Given a training set comprising $N$ examples $D_{\text{train}} = \{(x_i, y_i)\}_{i=1}^{N}$, we aim to learn a binary matching function $\mathcal{M}(x, y)$ that output high values for a paired sample $x$ and label $y$, while assigning lower values to unpaired ones. MEX parameterizes $\mathcal{M}(x, y)$ as $\text{Sim}(\mathcal{E}_s(x), \mathcal{E}_l(y))$, where $\mathcal{E}_s(\cdot) : \mathcal{X} \to \mathbb{R}^d$ represents the material encoder, $\mathcal{E}_l(\cdot) : \mathbb{R} \to \mathbb{R}^d$ represents the label encoder, and $\text{Sim}(\cdot, \cdot) : \mathbb{R}^d \times \mathbb{R}^d \to \mathbb{R}$ is the cosine similarity between two vectors. We denote the

encoded material and label as $\boldsymbol{z}^s$ and $\boldsymbol{z}^l$, respectively. Variables with subscript $i$ correspond to the $i$-th example in the training set.

The overall architecture of MEX is illustrated in Figure 2. In the following, we first outline the training process in Section 3.2.1, which aligns the sample and label feature spaces to capture their matching relationship from both absolute and relative perspectives. We then describe how to reformulate the regression to a matching problem in Section 3.2.2, which optimizes for the most matching label for a given sample based on the learned matching function.

### 3.2.1 TRAINING STAGE

In this section, we will introduce two training objectives for learning the sample-label matching relationship.

**Absolute matching optimization.** Since samples and labels are encoded separately, we optimize the cosine similarity between each $\boldsymbol{z}_i^s$ and $\boldsymbol{z}_i^l$ to align their feature spaces (left side, Figure 2-a), capturing the absolute material-property matching relationship:

$$\mathcal{L}_{abs,i} = -\frac{\boldsymbol{z}_i^s \cdot \boldsymbol{z}_i^l}{||\boldsymbol{z}_i^s|| \times ||\boldsymbol{z}_i^l||}, \tag{1}$$

and the total loss of $N$ samples is

$$\mathcal{L}_{abs} = \frac{1}{N} \sum_{i=1}^{N} \mathcal{L}_{abs,i}. \tag{2}$$

**Relative matching optimization.** Although $\mathcal{L}_{abs}$ enables the model to capture the matching relationship between a sample and its corresponding target value, it ignores relationships with other values, which is crucial for continuous label regression. To achieve this, we adopt the Noise Contrastive Estimation (NCE) (Gutmann & Hyvärinen, 2010) loss:

$$\mathcal{L}_{nce,i} = -\log \frac{\exp\left\{ \mathcal{S}\left(\boldsymbol{z}_i^s, \boldsymbol{z}_{(i,0)}^l\right) - \log q\left(y_{(i,0)} \mid y_i\right) \right\}}{\sum_{m=0}^{M} \exp\left\{ \mathcal{S}\left(\boldsymbol{z}_i^s, \boldsymbol{z}_{(i,m)}^l\right) - \log q\left(y_{(i,m)} \mid y_i\right) \right\}}, \tag{3}$$

where $\mathcal{S}(\cdot, \cdot) : \mathbb{R}^d \times \mathbb{R}^d \to \mathbb{R}$ refers to a non-linear score module (right side, Figure 2-a), which outputs the score of a sample representation and a label representation. We define $y_{(i,0)} := y_i$ and $\{y_{(i,m)}\}_{m=1}^{M}$ as $M$ noisy label values sampled from the noise distribution $q(y|y_i)$. Their corresponding label representations are denoted as $\{\boldsymbol{z}_{(i,m)}^l\}_{m=0}^{M}$. The noise distribution is modeled as a mixture of $K$ Gaussions centered at $y_i$ following Gustafsson et al. (2020):

$$q(y|y_i) = \frac{1}{K} \sum_{k=1}^{K} \mathcal{N}(y; y_i, \sigma_k^2). \tag{4}$$

The final NCE loss of the training samples is

$$\mathcal{L}_{nce} = \frac{1}{N} \sum_{i=1}^{N} \mathcal{L}_{nce,i}. \tag{5}$$

Minimizing $\mathcal{L}_{nce}$ encourages the model to distinguish the target value from noisy values, thereby capturing the fine-grained relative relationships between samples and labels more effectively.

By combining the absolute and relative matching optimization, the total training objective is:

$$\mathcal{L} = \mathcal{L}_{nce} + \lambda \mathcal{L}_{abs}, \tag{6}$$

where $\lambda$ is a trade-off parameter.

### 3.2.2 INFERENCE STAGE

During inference, the problem of predicting the target value of a sample $x$ can be formulated as finding a label $y^\star$ that best matches $x$. Based on the learned matching function, the prediction $y^\star$ can thus be obtained by directly maximizing the matching function $\mathcal{M}(x, y)$ w.r.t. $y$.

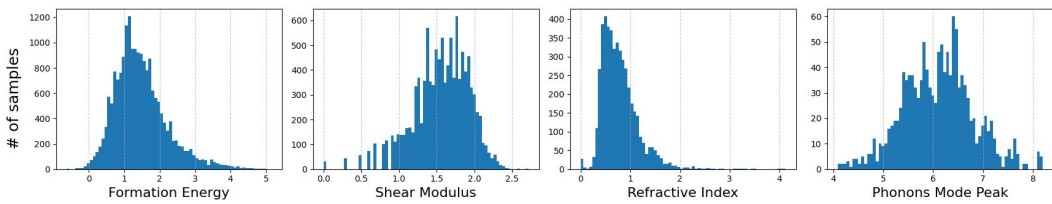

Figure 3: Overview of the label distribution for the origin MPR datasets. The X-axis denotes the respective property values. They were divided into seven benchmark datasets.

Table 1: Details of the seven benchmark datasets.

| Property | Num samples | Original source | Split configuration | Training label range | Val label range | Test label range |
|---|---|---|---|---|---|---|
| Formation Energy | 18982 | Castelli et al. (2012a) | bottom | [1.06, 5.16] | [0.76, 1.06] | [-0.64, 0.76] |
| Shear Modulus | 10987 | Jain et al. (2013) | bottom | [1.4, 2.72] | [1.18, 1.4] | [0, 1.18] |
| | | | top | [0, 1.78] | [1.78, 1.93] | [1.93, 2.72] |
| Refractive Index | 4764 | Petousis et al. (2017) | bottom | [0.56, 4.13] | [0.45, 0.56] | [0, 0.45] |
| | | | top | [0, 0.9] | [0.9, 1.11] | [1.11, 4.13] |
| Phonons Mode Peak | 1265 | Petretto et al. (2018) | bottom | [5.72, 8.2] | [5.41, 5.72] | [4.09, 5.41] |
| | | | top | [4.09, 6.45] | [6.45, 6.79] | [6.8, 8.2] |

We estimate $y^\star = \text{argmax}_y \mathcal{M}(x, y)$ with the Monte Carlo sampling-based stochastic optimization method (Homem-de Mello & Bayraksan, 2014), which iteratively refine a candidate label set $Y_{\text{cand}} = \{y_c\}_{c=1}^{C}$ based on probabilistic evaluations. This approach balances exploration and exploitation, allowing the matching values of candidate labels to converge toward high values. Finally, the prediction of $x$ is $y^\star = \text{argmax}_{y \in Y_{\text{cand}}} \mathcal{M}(x, y)$. The detailed algorithm for the inference procedure is provided in Appendix A.2.

## 4 BENCHMARKING EXTRAPOLATION IN MPR

### 4.1 DATASET

We curated a range of extrapolative MPR benchmark datasets using four datasets from Matminer (Ward et al., 2018) covering the following properties: formation energy, shear modulus, refractive index, and phonons mode peak, with dataset size ranging from 1,265 to 18,928. Figure 3 shows the label distribution of the raw datasets and more dataset characteristics are provided in Table 5. These datasets are split into train/validation/test sets with a ratio of 7:1.5:1.5. Rather than employing conventional partition methods like random splitting, we selected extreme target values for extrapolation evaluation. Specifically, each dataset is first sorted by the property values, and the top (or bottom) 15%, along with the second top (or bottom) 15%, were used as the test and validation sets, respectively. Whether we select the top or bottom extremes is determined by the specific property desired in material design scenarios. For example, lower formation energy indicates higher material stability, prompting researchers to search for materials with extremely low formation energies. Thus, for such properties, we select the bottom values for evaluation. For properties where both low and high values are of interest, e.g., shear modulus, the dataset is split with the top and bottom configuration once each to ensure comprehensive evaluation across the spectrum. Details of the resulting seven benchmark datasets are shown in Table 1.

### 4.2 BENCHMARK METHODS

**Network architectures.** We employ Geometric Graph Neural Networks (GNNs) (Han et al., 2024), which are designed to process data with geometric structures and have been widely used

Table 2: Test MAE(↓) on the benchmark dataset where BalancedMSE is abbreviated to BMSE. Bold is for the best and italics is for the second best in each column for both models. We report the standard deviation among 3 runs, consistent across all subsequent tables.

| Model | Algo | Formation Energy | Shear Modulus | | Refractive Index | | Phonons Mode Peak | | Avg Rank |
|---|---|---|---|---|---|---|---|---|---|
| | | bottom | bottom | top | bottom | top | bottom | top | |
| PaiNN | ERM | 0.424(0.001) | 0.613(0.089) | 0.363(0.000) | 0.275(0.007) | 0.781(0.017) | 0.820(0.005) | 0.975(0.022) | 6.4 |
| | LDS | 0.372(0.018) | 0.524(0.001) | 0.335(0.004) | 0.264(0.004) | 0.781(0.011) | 0.844(0.029) | 1.04(0.091) | 4.9 |
| | Ranksim | 0.421(0.002) | 0.540(0.001) | *0.246(0.030)* | 0.267(0.004) | 0.775(0.002) | *0.732(0.106)* | 1.00(0.071) | 4.4 |
| | BMSE | *0.360(0.050)* | **0.462(0.041)** | **0.214(0.038)** | 0.269(0.059) | *0.671(0.010)* | 0.758(0.011) | 1.02(0.018) | *3* |
| | ConR | 0.432(0.056) | 0.535(0.004) | 0.329(0.002) | 0.303(0.129) | 0.807(0.051) | 0.884(0.17) | 0.974(0.073) | 6.3 |
| | C-Mixup | 0.391(0.002) | 0.539(0.001) | 0.353(0.001) | 0.258(0.002) | 0.791(0.006) | 0.848(0.010) | *0.966(0.013)* | 5.1 |
| | FOMA | 0.419(0.005) | 0.502(0.005) | 0.351(0.043) | *0.239(0.046)* | 0.746(0.020) | 0.776(0.006) | 1.09(0.109) | 4.4 |
| | MEX | **0.309(0.018)** | *0.481(0.014)* | 0.298(0.008) | **0.177(0.019)** | **0.586(0.012)** | **0.567(0.008)** | **0.926(0.008)** | **1.4** |
| EquiformerV2 | ERM | 0.367(0.003) | 0.512(0.001) | 0.306(0.001) | 0.218(0.001) | 0.639(0.002) | 0.730(0.006) | 0.923(0.010) | 5.6 |
| | LDS | *0.278(0.008)* | 0.4944(0.004) | 0.295(0.005) | 0.195(0.009) | 0.643(0.002) | 0.749(0.001) | 0.905(0.012) | 3.9 |
| | Ranksim | 0.366(0.004) | 0.484(0.044) | 0.306(0.004) | 0.219(0.002) | 0.647(0.008) | 0.730(0.012) | 0.916(0.003) | 5.4 |
| | BMSE | 0.398(0.136) | *0.388(0.028)* | **0.167(0.020)** | *0.184(0.006)* | *0.599(0.013)* | *0.568(0.039)* | 1.02(0.022) | *3.4* |
| | ConR | 0.351(0.006) | 0.509(0.004) | 0.330(0.005) | 0.224(0.004) | 0.622(0.002) | 0.765(0.003) | *0.897(0.009)* | 5.6 |
| | C-Mixup | 0.316(0.009) | 0.509(0.001) | 0.319(0.003) | 0.205(0.003) | 0.628(0.008) | 0.752(0.002) | 0.915(0.005) | 5.1 |
| | FOMA | 0.314(0.004) | 0.512(0.004) | 0.311(0.009) | 0.196(0.004) | 0.627(0.002) | 0.768(0.020) | 1.068(0.222) | 5.9 |
| | MEX | **0.172(0.008)** | **0.376(0.010)** | *0.245(0.020)* | **0.141(0.004)** | **0.501(0.018)** | **0.495(0.007)** | **0.789(0.011)** | **1.1** |

Table 3: Test GM(↓) on the benchmark dataset where BalancedMSE is abbreviated to BMSE.

| Model | Algo | Formation Energy | Shear Modulus | | Refractive Index | | Phonons Mode Peak | | Avg Rank |
|---|---|---|---|---|---|---|---|---|---|
| | | bottom | bottom | top | bottom | top | bottom | top | |
| PaiNN | ERM | 0.388(0.001) | 0.571(0.094) | 0.340(0.001) | 0.228(0.008) | 0.709(0.020) | 0.750(0.006) | 0.849(0.020) | 6.7 |
| | LDS | 0.337(0.019) | 0.479(0.001) | 0.310(0.003) | *0.178(0.003)* | 0.668(0.008) | 0.770(0.026) | 0.910(0.098) | 4.6 |
| | Ranksim | 0.385(0.002) | 0.495(0.001) | *0.214(0.030)* | 0.223(0.008) | 0.659(0.0001) | 0.646(0.123) | 0.879(0.071) | 4.9 |
| | BMSE | *0.247(0.041)* | **0.367(0.041)** | **0.142(0.036)** | 0.205(0.074) | *0.501(0.010)* | *0.581(0.018)* | *0.800(0.021)* | *2* |
| | ConR | 0.385(0.076) | 0.488(0.005) | 0.306(0.002) | 0.275(0.144) | 0.737(0.061) | 0.817(0.182) | 0.845(0.082) | 6.1 |
| | C-Mixup | 0.354(0.003) | 0.493(0.001) | 0.330(0.002) | 0.213(0.002) | 0.689(0.200) | 0.781(0.012) | 0.852(0.014) | 5.6 |
| | FOMA | 0.383(0.005) | 0.454(0.006) | 0.331(0.045) | 0.203(0.052) | 0.634(0.019) | 0.701(0.008) | 0.979(0.120) | 4.7 |
| | MEX | **0.246(0.019)** | *0.432(0.016)* | 0.272(0.009) | **0.126(0.024)** | **0.495(0.013)** | **0.423(0.035)** | **0.756(0.031)** | **1.4** |
| EquiformerV2 | ERM | 0.330(0.003) | 0.466(0.001) | 0.288(0.001) | 0.172(0.001) | 0.550(0.003) | 0.667(0.008) | 0.829(0.007) | 5.9 |
| | LDS | *0.236(0.010)* | 0.446(0.004) | 0.275(0.004) | 0.136(0.0111) | 0.555(0.003) | 0.686(0.0011) | 0.806(0.0087) | 3.9 |
| | Ranksim | 0.329(0.004) | 0.436(0.049) | 0.288(0.004) | 0.172(0.002) | 0.561(0.010) | 0.669(0.014) | 0.825(0.006) | 5.7 |
| | BMSE | 0.301(0.134) | **0.298(0.032)** | **0.110(0.019)** | *0.106(0.001)* | *0.456(0.015)* | *0.389(0.026)* | 0.822(0.019) | *2.6* |
| | ConR | 0.318(0.005) | 0.465(0.004) | 0.310(0.004) | 0.180(0.005) | 0.527(0.001) | 0.704(0.003) | *0.800(0.010)* | 5.7 |
| | C-Mixup | 0.279(0.009) | 0.463(0.001) | 0.302(0.003) | 0.161(0.003) | 0.537(0.008) | 0.691(0.002) | 0.820(0.005) | 5 |
| | FOMA | 0.278(0.005) | 0.467(0.004) | 0.293(0.009) | 0.141(0.006) | 0.538(0.002) | 0.709(0.021) | 0.993(0.245) | 6 |
| | MEX | **0.113(0.006)** | *0.300(0.014)* | *0.208(0.025)* | **0.069(0.007)** | **0.364(0.030)** | **0.369(0.008)** | **0.631(0.014)** | **1.3** |

in material property prediction. We selected two representative equivariant Geometric GNNs: PaiNN (Schütt et al., 2021) and EquiformerV2 (Liao et al., 2024) from `fairchem`[1] as the backbone for all benchmark methods.

**Algorithms.** Given the limited number of proposals for extrapolation in the literature, we explore two categories of extrapolative regression methods. The first category is DIR methods. The second is the regression data augmentation (DA). To provide a comprehensive evaluation, we assess the performance of several representative methods from each category. Specifically, we choose LDS (Yang et al., 2021), Ranksim (Gong et al., 2022), BalancedMSE (Ren et al., 2022), and Conr (Keramati et al., 2024) for DIR methods; C-Mixup (Yao et al., 2022) and FOMA (Kaufman & Azencot, 2024) for regression DA. All these methods are benchmarked against the empirical risk minimization (ERM) baseline to evaluate their performance.

**Implementation details.** MEX is a general training framework agnostic to the material encoder. For the label encoder of MEX, we employ a linear layer attached by an activation function. The score module is a 4-layer Multi-layer Perceptron (MLP) that projects the concatenated sample and label representation to a score scalar. Besides, we empirically investigate various implementations of and compare their performance in Section 4.5.

---

[1] https://github.com/FAIR-Chem/fairchem?tab=readme-ov-file

In the training phase, 500 noisy labels are sampled for each example. We simply follow Gustafsson et al. (2020) to set $K = 3$ and $\sigma_1 = 0.075, \sigma_2 = 0.15, \sigma_3 = 0.3$ for the noisy distribution. During inference, the candidate label size is established at $C = 1500$, which is initially sampled uniformly from $[\lfloor l \rfloor, \lceil u \rceil]$, where $l$ and $u$ are the lower bound and upper bound of the entire dataset label range. Note that this interval can be freely adjusted based on prior knowledge of material properties. The candidate labels are updated for 10 iterations before we make the final prediction.

For all experiments, models were trained for a maximum of 200 epochs, with early stopping applied if the validation mean absolute error (MAE) did not improve for 30 consecutive epochs. We employed the AdamW (Loshchilov & Hutter, 2019) optimizer in conjunction with a `ReduceLROnPlateau` learning rate schedule, which reduced the learning rate by a factor of 0.8 after 5 epochs without improvement. Hyper-parameter selection was performed based on validation MAE via grid search, with the trade-off parameter $\lambda$ of MEX selected from $\{0.25, 0.5, 0.75, 1\}$, batch sizes from $\{32, 64, 128\}$, learning rates from $\{0.00005, 0.0001, 0.001\}$, and weight decay from $\{0, 0.001\}$. All methods were evaluated under three random seeds, and the average and standard deviation of MAE, error Geometric Mean (GM) (Yang et al., 2021), and Spearman correlation coefficient across all datasets were reported.

### 4.3 MAIN RESULTS

We report the performance for all methods in Table 2 and Table 3.

**MEX achieves superior extrapolation performance.** As shown in Table 2 and Table 3, MEX attains the best average rank across all models and both metrics, with the lowest MAE on 5 out of 7 datasets for PaiNN and 6 out of 7 for EquiformerV2. Under GM, MEX demonstrates the highest performance on 4 datasets for PaiNN and 5 for EquiformerV2. On other datasets, such as Shear Modulus, MEX performs competitively with the best-performing BalancedMSE method.

**DIR methods are strong baselines for extrapolation.** We observe that all DIR methods rank better than ERM on average for both models. For each dataset, at least one DIR method outperforms ERM, demonstrating their effectiveness for extrapolation. Notably, among the DIR methods, BalancedMSE consistently achieves the highest rank. However, no single DIR method outperforms ERM across all datasets, highlighting the need for methods specifically designed for extrapolation in MPR. Nevertheless, we recommend that future evaluations consistently include DIR methods as baselines due to their overall robustness.

**Regression DA helps extrapolation, but depends on models.** In Table 2, C-Mixup and FOMA consistently outperform ERM for PaiNN, with average ranks of 5.1 and 4.4, respectively, compared to 6.4 for ERM. However, their advantage diminishes when applied to EquiformerV2, where FOMA performs worse than ERM (rank 5.9 vs. 5.6). Although C-Mixup demonstrates better overall performance, the improvements on certain datasets, such as Shear Modulus (bottom), are marginal. A similar trend is observed under GM metrics (Table 3). This variability may arise from our application of augmentation in the feature space. Since different models produce latent representations of varying quality, the effectiveness of data augmentation methods fluctuates accordingly. This underscores the importance of developing robust material representation models, which are crucial for the success of feature-based data augmentation techniques. Additionally, it is interesting to design material-specific augmentation methods beyond those designed solely for continuous input data.

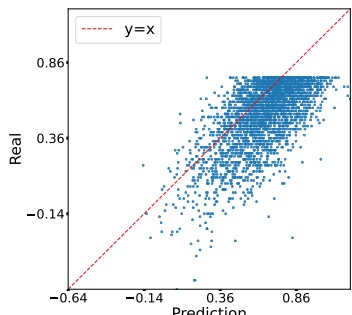

Figure 4: Prediction result of MEX on the test set of Formation Energy (bottom).

**Extrapolation remains a challenging problem.** The MAEs on our extrapolation benchmark are significantly larger than those obtained under random splits (Dunn et al., 2020). For instance, the early CGCNN (Xie & Grossman, 2018) achieves an MAE of 0.0452 (as reported by Dunn et al. (2020)) on the Formation Energy dataset under random split, which is considerably smaller than the smallest MAE (0.172, detailed results in Figure 4) under our extrapolative split configuration.

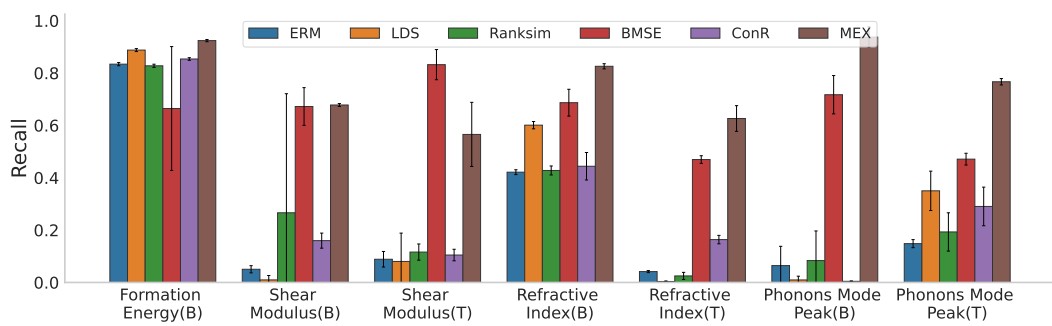

Figure 5: Recall rate of MEX and five DIR methods in detecting extrapolative samples.

To take a closer look at the prediction performance, we also calculate the Spearman correlation between the predictions and the target. We find that all methods exhibit weak (0-0.4) or even negative correlations with the targets across most datasets (quantization results are listed in Appendix Table 6). However, for the Formation Energy dataset, most methods achieve stronger correlations, which we hypothesize is due to the availability of sufficient data and the relative simplicity of the material structure in this task. In conclusion, accurate prediction for extrapolative samples remains extremely challenging for current methods.

## 4.4 POTENTIAL IMPACT ON CUTTING-EDGE MATERIAL DISCOVERY

As discussed in Section 4.3, extrapolation presents a significant challenge for methods in the literature, and our approach is no exception. Given the inherent limitations of neural networks in extrapolating (Xu et al., 2021), one may wonder: *To what extent can current deep learning methods assist in the discovery of cutting-edge materials?*

In addition to accurately predicting the property values of extrapolative samples, we contend that the ability to **detect** materials with potentially groundbreaking properties is also crucial. Once identified, these candidates can be further refined using first-principles methods, such as Density Functional Theory (DFT), to compute more precise properties. Consequently, models' detection capabilities could become vital tools in advancing material discovery.

To assess the effectiveness of different methods in detecting materials within extrapolation regions, we present their recall rates in Figure 5. Specifically, for samples from the validation and test sets in our benchmark, a sample is considered detected if its predicted value falls within the extrapolation interval, and the recall rate is calculated as the proportion of such samples correctly identified. As shown, MEX outperforms previous methods in 6 out of 7 detection tasks. Notably, it achieves a recall rate of over 80% on three datasets and exceeds 60% on six datasets. This substantial performance advantage demonstrates the robustness of MEX and highlights its potential to identify cutting-edge materials that might otherwise be overlooked.

## 4.5 DISCUSSION

**Score module analysis.** The score module is a critical component in learning fine-grained relationships between sample and label. Here, we investigate the effects of various design choices. The first, referred to as MEX (mlp+cos), employs two independent 2-layer MLPs to project the sample and label representations into a new space, after which the cosine similarity between the two projections is computed. The second approach, MEX (cos), directly computes the cosine similarity between the original sample and label representations. As illustrated in Table 4, MEX and MEX (mlp+cos) exhibit comparable performance, while MEX (cos) demonstrates inferior performance relative to the other designs. This observation aligns with findings in SimCLR (Chen et al., 2020), which indicate that incorporating a learnable nonlinear transformation on the representations before applying the contrastive loss, rather than directly optimizing the representations, significantly enhances the quality of the learned features.

Table 4: Test MAE and GM of EquiformerV2 on Formation Energy, Refeactive Index(bottom& top) datasets. MEX (cos) and MEX (mlp+cos) denote different designs of the score module in our framework.

| Metrics | MAE(↓) | | | GM(↓) | | |
|---|---|---|---|---|---|---|
| Dataset | Formation Energy | Refractive Index | | Formation Energy | Refractive Index | |
| | bottom | bottom | top | bottom | bottom | top |
| MEX (cos) | 0.382(0.004) | 0.231(0.003) | 0.625(0.007) | 0.346(0.004) | 0.184(0.004) | 0.533(0.005) |
| MEX (mlp+cos) | **0.169(0.014)** | 0.170(0.003) | 0.518(0.009) | **0.112(0.011)** | 0.100(0.005) | 0.373(0.013) |
| MEX | 0.172(0.008) | **0.141(0.004)** | **0.501(0.018)** | 0.113(0.006) | **0.069(0.007)** | **0.364(0.030)** |

**Trade-off parameter analysis.** We examine the selection of the trade-off parameter $\lambda$ by assessing model performance across various values of $\lambda$. Figure 6 illustrates the performance of MEX alongside prior top-performing methods on three benchmark datasets. As $\lambda$ changes, MEX consistently surpasses previous approaches across both models, thereby confirming its robustness to diverse hyperparameter configurations and backbone architecture choices.

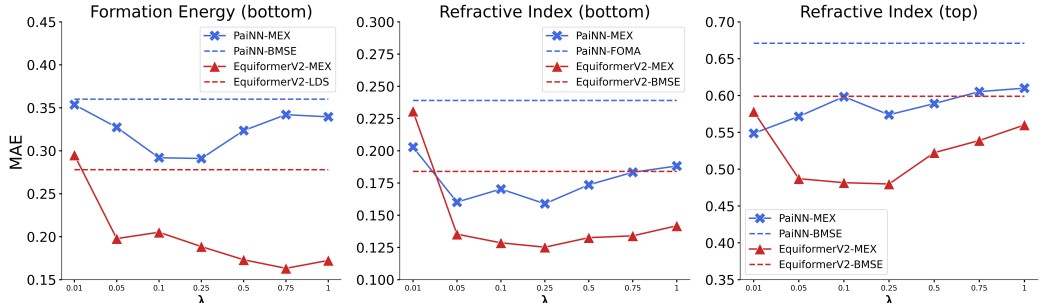

Figure 6: Ablation study on the trading-off parameter $\lambda$.

**Running time analysis.** Our method requires an iterative refinement of candidate labels before making the final prediction for each testing sample, which inherently results in a longer processing time compared to traditional regression methods. Specifically, this involves encoding 1,500 labels and computing their matching value over 10 iterations during our experiment. Despite this complexity, our experimental results indicate that the average computation time for MEX per test sample is about 0.006s on the NVIDIA 3090, which is comparable to baseline methods (around 0.002s). Thus, the computational overhead associated with our approach remains acceptable.

## 5 CONCLUSION

In this work, we shed light on the challenging task of extrapolation in material property regression (MPR), which aims to generalize to materials with unseen property values. We introduce a new benchmark consisting of seven MPR tasks and provide a comprehensive evaluation of existing methods' extrapolation capabilities. To address the task, we propose a simple yet effective framework that captures the sample-label matching relationship in the latent space. Extensive experiments demonstrate the superior performance of our approach and highlight its potential application in the discovery of cutting-edge materials.

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

# A APPENDIX

## A.1 DATASET DETAILS

Table 5: Dataset characteristics, including total atom types, atom numbers (mean and std.), and lattice constants (mean and std.). The symbols $a$, $b$, and $c$ denote the unit cell vectors. The notation $\|\cdot\|$ denotes the length of a vector and $\angle(\cdot,\cdot)$ denotes the angle between two vectors.

| Property | Atom Types | Atom Num. | $\|a\|$ | $\|b\|$ | $\|c\|$ | $\angle(b,c)$ | $\angle(a,c)$ | $\angle(a,b)$ |
|---|---|---|---|---|---|---|---|---|
| Formation Energy | 56 | 5 (0) | 4.14 (0.31) | 4.14 (0.31) | 4.14 (0.31) | 90.0 (0) | 90.0 (0) | 90.0 (0) |
| Shear Modulus | 84 | 8.63 (8.66) | 4.96 (1.5) | 5.33 (1.67) | 6.41 (2.98) | 83.29 (20.3) | 82.86 (19.78) | 85.35 (23.49) |
| Refractive Index | 80 | 16.9 (14.67) | 5.98 (1.94) | 6.6 (2.31) | 7.98 (3.61) | 86.32 (19.39) | 87.07 (19.12) | 89.55 (22.47) |
| Phonons Mode Peak | 64 | 7.53 (3.74) | 5.32 (1.42) | 5.66 (1.57) | 6.72 (2.09) | 83.55 (23.85) | 82.95 (23.4) | 84.1 (25.15) |

## A.2 INFERENCE ALGORITHM

---
**Algorithm 1:** Inference by Monte Carlo Sampling-Based Stochastic Optimization

---
**Input:** $x$: Input sample, $\mathcal{M}$: Matching function, $C$: Number of candidate labels, $T$: Iterations, $l$: Lower bound of label range, $u$: Upper bound of label range, $\beta$: noise shrink factor

**Output:** $y^\star$: Optimal label

$ns \leftarrow$ initial noise scale;

$\{y_i \sim \mathcal{U}(l,u)\}_{i=1}^C \leftarrow$ initial labels;

// uniform sample from $[l,u]$

**for** $t \leftarrow 1$ **to** $T$ **do**

    $\{p_i\}_{i=1}^C \leftarrow \text{Softmax}(\{\mathcal{M}(x,y_i)\}_{i=1}^C)$

    $\{y_i\}_{i=1}^C \leftarrow \text{sample}(\{y_i\}_{i=1}^C, \{p_i\}_{i=1}^C)$;

    // Sampling based on probability with replacement

    **for** $i \leftarrow 1$ **to** $C$ **do**

        $\epsilon_i \sim \mathcal{N}(0,1)$;

        $y_i \leftarrow y_i + \epsilon_i * ns$;

        $y_i \leftarrow \text{clip}(y_i, l, u)$;

        // clip $y_i$ to $[l,u]$

    **end**

    $ns \leftarrow \beta * ns$;

    // shrink noise scale

**end**

$y^\star \leftarrow \text{argmax}_{y \in \{y_i\}_{i=1}^C} \mathcal{M}(x,y)$;

**return** $y^\star$;

---

## A.3 EXPERIMENT DETAILS

### A.3.1 EVALUATION METRICS

**MAE.** Mean Absolute Error (MAE) is defined as $\frac{1}{N}\sum_{i=0}^{N}|y_i - \hat{y}_i|$, where $N$ is the number of samples. $y_i$ and $\hat{y}_i$ are the ground truth label and prediction of the $i$-th sample, respectively. Lower is better.

**GM.** Error Geometric Mean (GM) is defined as $(\Pi_{i=0}^{N}|y_i - \hat{y}_i|)^{1/N}$, where $N$ is the number of samples. $y_i$ and $\hat{y}_i$ are the ground truth label and prediction of the $i$-th sample, respectively. Lower is better. We implement GM as $\left(\Pi_{i=0}^{N}\max\{|y_i - \hat{y}_i|, 10^{-10}\}\right)^{1/N}$ for metric robustness.

**Spearman correlation.** Spearman correlation measures the direction of the monotonic relationship between two variables by calculating the Pearson correlation on their ranked values. We use the implementation in `scipy` library. Higher is better.

### A.3.2 SPEARMAN CORRELATION FOR ALL METHODS

Table 6: Test Spearman correlation efficient on the benchmark dataset where BalancedMSE is abbreviated to BMSE.

| Model | Algo | Formation Energy | Shear Modulus | | Refractive Index | | Phonons Mode Peak | |
|---|---|---|---|---|---|---|---|---|
| | | bottom | bottom | top | bottom | top | bottom | top |
| PaiNN | ERM | 0.541(0.005) | 0.059(0.148) | -0.128(0.047) | -0.116(0.039) | -0.208(0.065) | **-0.181(0.007)** | -0.421(0.004) |
| | LDS | **0.660(0.013)** | 0.171(0.028) | -0.022(0.080) | -0.104(0.024) | -0.265(0.018) | -0.230(0.043) | -0.379(0.018) |
| | Ranksim | 0.542(0.004) | 0.170(0.017) | -0.018(0.023) | -0.070(0.0430) | -0.314(0.009) | -0.216(0.046) | -0.418(0.012) |
| | BMSE | 0.351(0.044) | 0.099(0.064) | -0.054(0.055) | -0.133(0.066) | -0.237(0.022) | -0.260(0.054) | **-0.302(0.017)** |
| | ConR | 0.473(0.101) | 0.131(0.019) | -0.032(0.033) | 0.009(0.029) | -0.174(0.036) | -0.210(0.029) | -0.430(0.018) |
| | C-Mixup | 0.567(0.005) | 0.144(0.019) | -0.093(0.017) | -0.052(0.016) | -0.213(0.129) | **-0.181(0.010)** | -0.437(0.014) |
| | FOMA | 0.534(0.013) | 0.232(0.020) | -0.046(0.080) | **0.093(0.018)** | -0.325(0.005) | -0.205(0.044) | -0.384(0.042) |
| | MEX | 0.489(0.051) | **0.319(0.015)** | **0.042(0.003)** | 0.059(0.027) | **0.039(0.039)** | -0.201(0.038) | -0.335(0.018) |
| EquiformerV2 | ERM | 0.615(0.015) | 0.336(0.042) | 0.069(0.024) | 0.194(0.015) | -0.120(0.016) | 0.008(0.054) | -0.459(0.011) |
| | LDS | 0.71(0.044) | 0.155(0.033) | 0.020(0.085) | **0.237(0.020)** | -0.031(0.013) | 0.009(0.117) | -0.411(0.030) |
| | Ranksim | 0.625(0.014) | 0.332(0.009) | 0.094(0.016) | 0.195(0.021) | -0.060(0.004) | **0.061(0.071)** | -0.432(0.015) |
| | BMSE | 0.273(0.043) | 0.278(0.033) | 0.074(0.020) | -0.016(0.004) | -0.126(0.0470) | -0.175(0.037) | -0.393(0.003) |
| | ConR | **0.724(0.012)** | 0.357(0.023) | 0.109(0.038) | -0.021(0.047) | -0.022(0.008) | -0.050(0.108) | -0.460(0.047) |
| | C-Mixup | 0.682(0.019) | 0.328(0.063) | 0.117(0.055) | 0.183(0.077) | -0.045(0.014) | 0.005(0.003) | -0.479(0.008) |
| | FOMA | 0.703(0.004) | 0.317(0.032) | **0.131(0.060)** | 0.211(0.028) | -0.012(0.003) | 0.040(0.042) | **-0.360(0.184)** |
| | MEX | 0.645(0.020) | **0.374(0.021)** | 0.095(0.027) | 0.080(0.027) | **0.088(0.006)** | -0.039(0.085) | -0.427(0.019) |

