# OpenReview forum: "ON EXTRAPOLATION IN MATERIAL PROPERTY REGRESSION"
_ICLR.cc/2025/Conference — Submitted to ICLR 2025_

### Official Review · Reviewer_N2Xe · 2024-10-30

**Soundness:** 2
**Presentation:** 2
**Contribution:** 1
**Rating:** 3
**Confidence:** 2

**Summary:**

The author addresses the issue of material property regression, focusing specifically on regression problems where the feature values lie outside the boundaries of the training set.
The author performs the regression task using two encoder models: (1) an encoder for material properties and (2) an encoder for target values.
Finally, using Monte Carlo sampling, the model outputs target values that can yield features most similar to the object properties within the given boundary.
The author structured the dataset using Matminer ensuring that the target values in the training and evaluation environments do not overlap.
The proposed method achieved a high level of performance compared to other methods.

**Strengths:**

The problem setting proposed by the author is, to some extent, justifiable, and within this setting, the author presents a state-of-the-art algorithm.

**Weaknesses:**

(1) Narrowly defined problem:
The experimental setting proposed by the author is highly narrow in scope.
An algorithm that performs well only within the proposed setting does not provide insight into whether it effectively considers material properties within the boundaries of the training data.
Furthermore, the experimental setting proposed by the author appears highly challenging, and the actual MAE values are relatively large compared to the target values.
Therefore, it is difficult to conclude that the author has sufficiently demonstrated the practicality of the proposed method.

(2) Main method which lacks novelty and analysis:
Specifically, it is challenging to discern any reasonable approach proposed by the author for addressing extrapolation.
The two matching optimizations proposed by the author are both methods aimed at accurately predicting the given labels in the training set.
The author utilizes MC sampling to identify target values outside the range of the training set; however, it is unclear whether the encoding of target values beyond the training range has been effectively learned.
This suggests that the algorithm "may" achieve high performance if there is significant variance in the value encoder, hyperparameters, and other factors.
Therefore, to verify the extrapolation performance of the proposed method, the author should present additional ablation studies.

(3) Lack of fairness:
First, there is a question regarding the author’s validation setting.
It is unclear why the target value range is the same in both the test and validation environments.
Second, the author is aware of the lower and upper bounds of the label range but does not address the issue of setting these bounds beyond the range of the test set.
This strongly conflicts with the motivation for extrapolation that the author discusses.
Lastly, the author does not provide the hyperparameter search space for the comparative methods.
These are details that should be explicitly documented if new training was conducted on a new dataset.

**Questions:**

The first question concerns whether the practicality of the author’s method can be extended.
Naturally, defining the boundary of target values in the training set is highly unnatural.
It is natural for materials with target values similar to those of a new material to be sparsely represented.
This implies that the author’s algorithm should also be capable of addressing imbalanced regression problems.
Can the author demonstrate the effectiveness of this algorithm on established imbalanced regression benchmarks or in environments where labels are sparsely shared between the training and evaluation sets?
If feasible, the author should suggest providing a rigorous comparison framework to ensure fairness, as discussed in the weaknesses section.

The second question is whether the author can explain how the method intuitively aligns with the concept of extrapolation.
Whether through theoretical or experimental approaches, it is essential to establish confidence that this method genuinely addresses the MPR problem with consideration for extrapolation.

---

> ### Author Response · Authors · 2024-11-19
> **To Reviewer N2Xe (Part Ⅰ)**
>
> We thank the reviewer for the rigorous reviews. We have carefully and thoroughly addressed the concerns and sincerely hope the reviewer will consider raising the evaluation score.
>
> **Response to weaknesses**
> > 1.1: About the problem scope.
>
> A: This work focuses on the critical challenge of extrapolation in MPR—predicting property values beyond the training data, which is essential for AI-powered material discovery with novel properties. The i.i.d tests suggested by the reviewer has been well-studied and is not the focus of this paper.
>
> > 1.2: About the practicality of the proposed method.
>
> A: In Section 4.4, we have demonstrated that our method can effectively recall materials with novel properties, highlighting its practical value for material discovery.
>
> > 2.1:About method novelty
>
> A: We would like to emphasize that novelty is not limited to technical advancements but can take various forms. In our work, we reframe MPR as a material-property matching task specifically tailored for extrapolation, introducing a novel application of matching techniques to this critical challenge and offering fresh insights to the field of MPR.
>
> > 2.2: About the rationality of our method for addressing extrapolation.
>
> A: While the proposed matching optimizations focus on modeling the relationship between x and y within the training set, this allows the model to learn the compatibility between x and y, enabling it to infer the compatibility between unseen x and unseen y.  This neural network capability to generalize beyond the training data is not unique to our approach but rather reflects a broader principle observed in related domains, such as zero-shot learning (where the model is trained solely on seen images and classes, then tested on unseen images and classes) and information retrieval (where the model is trained solely on seen queries and documents, then tested on unseen queries and documents).
>
> > 2.3:  the algorithm "may" achieve high performance if there is significant variance in the value encoder, hyperparameters, and other factors.
>
> A: As shown in Table 2 and 3, our method outperforms existing methods under a wide range of datasets, backbones and metrics, demonstrating its robustness. Although we conducted hyperparameter searches, these were based on performance on the validation set, not the test set. For fairness, we also report the results of hyperparameter searches for the baselines below (in 3.3), providing additional evidence of our method’s superiority. If the reviewer has specific suggestions for additional ablation studies, we would be glad to conduct further experiments.
>
> > 3.1:  It is unclear why the target value range is the same in both the test and validation environments.
>
> A: As provided in Section 4.1, we clarify that the target value ranges of the validation and test sets are indeed **disjoint** across all seven tasks. In Table 1, we merged the validation and test ranges for brevity, which may have caused your misunderstanding. We have uploaded a newer revision, which displays the label ranges for training, validation, and test sets separately in Table 1.
>
> > 3.2: About the lower and upper bounds of the label range.
>
> A: To address the reviewer's concern, we extend the bounds from dataset-specific values to broader ones, namely [-10, 10] for all datasets, which can encompass the theoretical range for most materials-related properties. As shown in the following table, even with these broader bounds, our method maintains superior performance, demonstrating its robustness to bound settings.
>
> |  Metric  | Search Interval   | Formation Energy (bottom) | Shear Modulus (bottom) | Shear Modulus (top) | Refractive Index (bottom) | Refractive Index (top) | Phonons Mode Peak (bottom) | Phonons Mode Peak (top) | Avg. Rank |
> |-|-|-|-|-|-|-|-|-|-|
> | MAE        | orgin      | 0.172(0.008)              | 0.376(0.010)           | 0.245(0.020)        | 0.141(0.004)            | 0.501(0.018)           | 0.495(0.007)               | 0.789(0.011)           | 1.1 |
> |            | [-10, 10]  | 0.176(0.004)              | 0.407(0.008)           | 0.248(0.017)        | 0.133(0.002)            | 0.519(0.011)           | 0.509(0.002)               | 0.809(0.018)           | 1.3 |
> | GM         | orgin      | 0.113(0.006)              | 0.300(0.014)           | 0.208(0.025)        | 0.069(0.007)            | 0.364(0.030)           | 0.369(0.008)               | 0.631(0.014)           | 1.3 |
> |            | [-10, 10]  | 0.117(0.002)              | 0.338(0.011)           | 0.201(0.013)        | 0.057(0.00)             | 0.381(0.02)            | 0.373(0.015)               | 0.6674(0.022)          | 1.3 |

---

> ### Author Response · Authors · 2024-11-19
> **To Reviewer N2Xe (Part Ⅱ)**
>
> > 3.3: About the hyperparameter search space for the comparative methods.
>
> A: We followed the reviewer's suggestion and conducted hyperparameter searches for the baselines. As shown in the table below, our method still achieves SOTA performance in 12 out of 14 comparisons. Specifically, we performed task-agnostic hyperparameter searches, such as learning rate optimization, as detailed in Section 4.2. For task-specific hyperparameters, we identified that Ranksim, ConR, C-Mixup, and FOMA include key settings that significantly influence performance. We adhered to the hyperparameter search protocols outlined in their respective papers and reported the best MAEs in the table.
>
> | Model          | Algorithm | Formation Energy (bottom) | Shear Modulus (bottom) | Shear Modulus (top) | Refractive Index (bottom) | Refractive Index (top) | Phonons Mode Peak (bottom) | Phonons Mode Peak (top) |
> |-|-|-|-|-|-|-|-|-|
> | PaiNN          | ConR      | 0.403(0.012)             | 0.535(0.004)           | 0.329(0.002)       | 0.303(0.129)             | 0.74(0.06)             | 0.772(0.015)               | 0.939(0.007)           |
> |                | Ranksim   | 0.42(0.003)              | 0.54(0.001)            | **0.246(0.03)**        | 0.267(0.004)             | 0.775(0.002)           | 0.732(0.106)           | 0.983(0.046)       |
> |                | C-Mixup   | 0.387(0.005)             | 0.537(0.002)           | 0.349(0.001)       | 0.257(0.005)             | 0.788(0.004)           | 0.82(0.008)                | 0.966(0.013)           |
> |                | FOMA      | 0.401(0.004)             | **0.446(0.056)**       | 0.312(0.071)       | 0.219(0.025)             | 0.746(0.02)            | 0.776(0.006)               | 0.991(0.024)           |
> |                |MEX        |**0.309(0.018)** |0.481(0.014) |0.298(0.008) |**0.177(0.019)** | **0.586(0.012)** | **0.567(0.008)** | **0.926(0.008)**|
> | EquiformerV2   | ConR      | 0.351(0.006)             | 0.509(0.004)           | 0.326(0.006)       | 0.222(0.004)             | 0.621(0.006)           | 0.735(0.004)               | 0.897(0.009)           |
> |                | Ranksim   | 0.356(0.003)             | 0.467(0.075)           | 0.304(0.003)       | 0.217(0.002)             | 0.624(0.002)           | 0.727(0.005)               | 0.915(0.005)           |
> |                | C-Mixup   | 0.312(0.016)             | 0.509(0.001)           | 0.314(0.002)       | 0.205(0.003)             | 0.626(0.003)           | 0.752(0.002)               | 0.915(0.005)           |
> |                | FOMA      | 0.314(0.004)             | 0.511(0.001)           | 0.311(0.001)       | 0.196(0.004)             | 0.627(0.002)           | 0.741(0.007)               | 0.914(0.001)           |
> |                | MEX       |**0.172(0.008)** |**0.376(0.010)** |**0.245 (0.020)** |**0.141(0.004)** |**0.501(0.018)** |**0.495(0.007)** |**0.789(0.011)**|
>
> **Response to questions**
>
> > Q1: Can the author demonstrate the effectiveness of this algorithm on established imbalanced regression benchmarks or in environments where labels are sparsely shared between the training and evaluation sets?
>
> A: We would like to emphasize that our benchmark, although challenging, is significant for material discovery. To solve the reviewer's concern, we construct a setting where training labels overlap slightly with the target. Specifically, we randomly chose 10 samples from the validation and test set separately and moved them to the training set, forming an imbalanced regression scenario. We compared our method and SOTA baseline methods in the following table. Both methods exhibit a performance improvement compared to the original setting and our method is still effective in most tasks.
> |Metric     | Algorithm  | Formation Energy (bottom) | Shear Modulus (bottom) | Shear Modulus (top) | Refractive Index (bottom) | Refractive Index (top) |Phonons Mode Peak (bottom) |Phonons Mode Peak (top)
> |:--|:--|:--|:--|:--|:--|:--|:--|:--|
> |MAE($\downarrow$)|SOTA Baseline|0.228(0.008) |0.364(0.017) |**0.181(0.021)** |0.190(0.012) |0.553(0.01) |0.417(0.012) |**0.331(0.035)** |
> |                 |MEX          |**0.162(0.01)** |**0.356(0.003)** |0.236(0.012) |**0.136(0.006)** |**0.531(0.008)** |**0.414(0.023)** |0.362(0.021)|
> |GM($\downarrow$)|SOTA Baseline |0.179(0.009) |0.277(0.023) |**0.125(0.026)** |0.105(0.01) |0.406(0.016) |**0.283(0.012)** |0.22(0.036)  |
> |                 |MEX          |**0.106(0.008)** |**0.265(0.006)** |0.183(0.01) |**0.062(0.003)** |**0.388(0.027)** |0.306(0.035) |**0.219(0.02)**  |
>
> > Q2: how the method intuitively aligns with the concept of extrapolation.
>
> A: - This question has been addressed in weakness 2.2.

---

> ### Author Response · Authors · 2024-11-22
>
> Dear Reviewer N2Xe,
>
> Thank you once again for your valuable and constructive feedback on our submission. As the discussion period progresses toward its end, we kindly request any additional comments or thoughts you may have regarding the clarifications and results we have provided in our rebuttal. We fully understand that this is a busy time, and we sincerely appreciate the effort and time you have dedicated to reviewing our work. If there are any remaining questions or concerns, we would be happy to address them promptly.
>
> Best regards, The Authors

---

> ### Author Response · Authors · 2024-11-25
> **Look forward to your post-rebuttal feedback!**
>
> Dear Reviewer N2Xe,
>
> Thanks again for your insightful suggestions and comments. Since the deadline of discussion is approaching, we are happy to provide any additional clarification that you may need.
> In our previous response, we have carefully studied your comments and made detailed responses summarized below:
> - Provide further explanation of the novelty, practicality and rationality of our method.
> - Clarify the label range of the validation set to ensure experiment fairness.
> - Conduct additional experiments (dataset-agnostic label interval for MEX & hyperparameter search for baselines) to ensure fairness of the comparisons.
> - Perform additional experiments to demonstrate the effectiveness of our method on imbalanced MPR.
>
> We hope that the explanation of our method and the provided new experiments have convinced you of the contributions of our submission.
>
> Please do not hesitate to contact us if there's additional clarification or experiments we can offer. Thanks!
>
> Thank you for your time!
>
> Best, Authors

---

> ### Comment · Reviewer_N2Xe · 2024-11-26
>
> Thank you for the detailed explanations provided by the authors.
> Below, I have outlined the remaining concerns.
> In summary, I believe that the proposed MPR with extrapolation (MPRE) problem could serve as a valuable contribution to the community, provided that it is both (1) practical and (2) solvable.
>
> 1. Practicality (about (1))
>
> The authors claim that solving the MPRE problem can aid in material discovery. However, the experiment in Section 4.4 resembles an anomaly detection task and does not report precision metrics.
> Do the authors have examples of applications where the proposed method can be used outside the MPRE context?
> If the authors can provide such examples, the proposed MPRE problem has the potential to serve as a benchmark dataset. Without this, the concern remains that the problem may be too narrowly defined.
>
> 2. Solvability (about (1) and (2))
>
> I am curious whether the proposed MPRE problem is indeed solvable.
> While I appreciate the experiments where the authors added 10 samples to the training data, this does not fully address my concerns.
> Even with just 10 additional samples, the MAE values decreased significantly.
> As mentioned in my initial review, the reported MAE values appear high relative to the target value.
> Are the authors confident that MPRE is a solvable problem? Additionally, does the proposed setting (using 70% of the training data) have physical or scientific justification?
>
> 3. Fairness (about (2))
>
> Could the authors provide more details about the validation setting?
> Specifically, I am curious whether the model is retrained on a dataset that includes the validation data after hyperparameter tuning.
>
> 4. Method Effectiveness (about (2))
>
> I understand that the label encoder approach proposed by the authors is related to extrapolation. However, it is well-known that over-parameterized deep learning models often lead to overfitting.
> Additionally, the NCE loss proposed by the authors appears to focus on reducing confidence around the labels, making the model more robust to noise (as evidenced by improved extrapolation performance with larger lambda values).
> Have the authors considered methods to mitigate overfitting within the target range of the training data, particularly to ensure effective extrapolation?
> I suspect the validation setting plays a crucial role here, and I am curious whether early stopping based on validation performance is a key factor in the effectiveness of the proposed approach.

---

> ### Author Response · Authors · 2024-11-27
> **To Reviewer N2Xe (Part Ⅲ)**
>
> Thank you for your insightful feedback. We have provided detailed explanations for each question below and hope these address your concerns.
>
> > Q1.1: about the precision metric.
>
> A: The detection task in Section 4.4 evaluates whether an extrapolative sample (label 1) is correctly classified as extrapolative (label 1) or not (label 0). In the test set, all samples are extrapolative (label 1), meaning there are no false positives. As a result, the precision is always 1 for all methods in this setting, rendering it uninformative for comparison.
>
> > Q1.2: about more applications of MEX
>
> A: First, while our focus is on extrapolation related to material properties, we do not consider this a narrowly defined problem. Materials are fundamental to the development of human society and are deeply interconnected with critical domains such as energy and the environment. For instance, breakthroughs in discovering innovative catalysts for renewable energy storage could significantly advance efforts to combat climate change. Moreover, we believe that other types of scientific data [1,2], including small-molecule drugs, proteins, and genetic information, also face similar extrapolation challenges.  In the future, we will explore more possibilities of MEX in these areas.
>
> > Q2.1: about the solvability of the problem.
>
> A: We firmly believe that MPRE is a theoretically solvable problem. This is because the mapping between a material’s structure and its properties is governed by fundamental physical laws (e.g., quantum mechanics at the atomic scale). Regardless of the material distribution, these physical principles **remain invariant**. Therefore, the better a model is at capturing these robust principles, the better its extrapolation performance will be. Moreover, recent work [3] in machine learning has shown theoretically that encoding proper non-linearities can enable effective extrapolation, further supporting the feasibility of solving MPRE.
>
> > Q2.2: about the data splitting.
>
> A: The 70%-15%-15% split for training, validation, and testing is a widely adopted practice in the machine learning community, and material property prediction tasks are no exception [4].
>
> > Q3.1: about the validation setting.
>
> A: Details of the validation setting are explicitly provided in Section 4.1 and Table 1. The validation set is entirely disjoint from both the training and test sets, specifically designed to mimic extrapolation scenarios during training. This design aligns with standard practices in areas like domain generalization (where train/val/test lie in different domains) and zero-shot learning (where train/val/test involve disjoint classes). If the reviewer has specific concerns, we would be happy to address them.
>
> > Q3.2: training with validation data.
>
> A: We strictly adhere to the standard train-validation-test pipeline. The model is never retrained on datasets that include validation data.
>
> > Q4.1: about overfitting.
>
> A: Experimental results presented in Section 4.3 and Section 4.4 indicate that our methods exhibit minimal overfitting within the training label range. For example, as discussed in Section 4.4, MEX is less likely to mistake extrapolative labels as within the training label interval. While our method is an initial attempt to address the MPRE challenge, we believe it opens the door for future improvements in this area.
>
> > Q4.2: about early stopping.
>
> A: Early stopping based on validation performance is standard practice in machine learning to avoid overfitting, and we consistently applied this technique across all methods during our experiments.
>
> **References**
>
> [1] Chan, Alvin, et al. "Deep extrapolation for attribute-enhanced generation."
>
> [2] Padmakumar, Vishakh, et al. "Extrapolative Controlled Sequence Generation via Iterative Refinement."
>
> [3] Xu, Keyulu, et al. "How neural networks extrapolate: From feedforward to graph neural networks."
>
> [4] Chang, Rees, et al. "Towards Overcoming Data Scarcity in Materials Science: Unifying Models and Datasets with a Mixture of Experts Framework."

---

> ### Comment · Reviewer_N2Xe · 2024-11-27
>
> For Q1.1
>
> The authors should also evaluate the algorithm on in-range samples to measure precision.
>
> For Q1.2
>
> The authors have designed the MPRE problem under the assumption of an extreme extrapolation scenario.
> The reason I argue that the problem is narrowly defined is that the authors do not consider regression for in-range samples at all
> (This concern was partially resolved by the additional exp.).
> As stated in the authors' response to Q2.1, if immutable physical principles exist and it is possible to learn these principles from in-range samples, there is no reason not to test the model on in-range samples.
> The problem proposed by the authors is both interesting and necessary; however, as the authors are introducing a new benchmark, it is essential to provide robust justification for its validity.
>
> For Q2.1 & Q2.2
>
> Unfortunately, the paper [4] does not partition the data based on the target value.
> Contrary to the authors' opinion, I am not convinced that deep learning models can reliably learn physical rules invariant to the range of target values (referred to by the authors as the material distribution).
> Indeed, the additional experimental results provided by the authors in their rebuttal also demonstrate a sharp improvement in model performance when additional samples are utilized.
>
> For Q3
>
> The proposed method appears to have a higher dependency on the validation set compared to other approaches.
> This suggests that the presence or absence of a validation set (for hyperparameter tuning) could have a significant impact on the performance of the method.
> The authors could identify the hyperparameters using the validation set and then retrain the model on a dataset that includes the validation set using the identified hyperparameters (see [R-1]).
> Alternatively, the authors could experimentally demonstrate that the validation set plays a minimal role in the proposed method.
>
> For Q4
>
> Is it correct that the proposed method relies entirely on early stopping as a tool to prevent overfitting?
> If so, the effectiveness of early stopping should also be thoroughly discussed in the paper.
>
> [R-1] Erik Englesson, et al. "Generalized Jensen-Shannon Divergence Loss for Learning with Noisy Labels"

---

> > ### Comment · Reviewer_N2Xe · 2024-11-27
> >
> > I apologize for the somewhat direct nature of my review, and I also sincerely thank the authors for their kind and detailed rebuttal.
> >
> > Despite multiple discussions, I remain unconvinced that MPRE is not a narrowly defined problem.
> > As Reviewer oJTR noted, proposing a new benchmark is both a challenging and valuable endeavor.
> > A benchmark that will continue to be referenced by the community must be proposed with rigor and robustness.
> >
> > The authors’ efforts to address concerns about practicality [1,2] and justification for the dataset construction [4] have alleviated some of my concerns, leading me to revise my initial rating upward.

---

### Official Review · Reviewer_JB71 · 2024-10-31

**Soundness:** 3
**Presentation:** 3
**Contribution:** 3
**Rating:** 5
**Confidence:** 4

**Summary:**

This paper explores the challenge of extrapolation in material property regression. Existing deep learning models for MPR assume that the training and test data follow similar distributions, thus limiting their ability to make predictions beyond the known range. To address this issue, the authors introduce the Matching-based Extrapolation (MEX) framework that reframes MPR as a material-attribute matching problem.MEX employs both absolute and relative matching objectives to optimize the consistency of the material and attribute representations, thereby facilitating better extrapolation of material property predictions. The authors also develop a new benchmark.

**Strengths:**

- Originality:
  - Novel perspective of material property prediction. Reframe the task as a material property matching problem.
- Quality:
  - A new benchmark is constructed and a new framework is proposed to address the critical problem of out-of-domain material property prediction.
- Clarity:
  - Most of the paper is clearly presented, with some details that should be explained more clearly. The section on noise contrastive estimation-based optimization.
- Significance:
  - The problem is critical for new material discovery. The benchmark provides an important validation of the OOD material property prediction problem.

**Weaknesses:**

- Limited novelty in method.
  - The key components, NCE, and cosine similarity-based matching are well-known techniques.
- scalability and computational complexity
  - The MEX framework’s inference requires iterative candidate label refinement, which introduces considerable computational overhead compared to traditional regression methods.
- Experiments
  - Traditional methods, like DFT, are supposed to be compared to find the gap between DL-based extrapolation and traditional methods.
  - More DL-based methods should be compared to provide evidence that previous works lack the generalization of OOD properties.

**Questions:**

- Dataset
  - The diversity of the dataset, for example, the distribution of atom numbers and the lattice constants, etc.
- Matching choice
  - Are there any insights into adapting Noise Contrastive Estimation (NCE) instead of other methods?

---

> ### Author Response · Authors · 2024-11-19
> **To Reviewer JB71**
>
> We appreciate the reviewer for acknowledging our studied problem as significant and our perspective of solving extrapolation as novel. We sincerely believe that we have effectively addressed the reviewer’s main concerns and hope that the reviewer may consider raising the evaluation score to reflect the improvements made in our paper.
>
> **Response to weaknesses**
>
> > 1: About method novelty.
>
> A: We would like to emphasize that novelty is not limited to technical advancements but can take various forms. In our work, we reframe MPR as a material-property matching task specifically tailored for extrapolation, introducing a novel application of matching techniques to this critical challenge and offering fresh insights to the field of MPR.
>
> > 2: scalability and computational complexity.
>
> A:   First, in Section 4.5, we delve into the runtime performance of the MEX framework, illustrating that its computational efficiency is comparable to conventional regression techniques. Additionally, we demonstrate that the computational expense can be further reduced without sacrificing performance by tuning two key hyperparameters: the number of candidate labels (C) and the number of optimization iterations (T). Taking the Formation Energy task as a case study, we adjusted C to 500 and T to 3, reducing the processing time to 0.0031 seconds per sample, which is on par with the baseline LDS.
> |     | Running time (second per sample)  |MAE($\downarrow$) |GM($\downarrow$)|
> |:----------|:---------|:-----------|:-----------|
> |LDS (baseline)|0.0025 |0.288 |0.2439 |
> |MEX (C=1500, T=10, original setting) |0.0061 | 0.1764| 0.1155 |
> |MEX (C=500, T=3)|0.0031 |0.1762 |0.116 |
>
> > 3.1:  Traditional methods, like DFT, are supposed to be compared to find the gap between DL-based extrapolation and traditional methods.
>
> A:  DFT is not directly compared here as as the dataset was annotated using it.
>
> >3.2: more DL-based methods should be compared
>
> A: As extrapolation remains an underexplored area, there are currently few established methods. Although earlier studies [1,2,3] examined extrapolation, they were limited to simplified experiments without practical methodologies.  To the best of our knowledge, DIR methods could serve as competitive baselines. For instance, LDS and Ranksim, included in our experiments, claim effectiveness in this domain and demonstrate strong performance on disjoint label ranges. If the reviewer has additional suggestions for suitable baselines, we will make every effort to include them and provide experimental results.
>
> **Response to questions**
>
> > Q1: About the diversity of the dataset.
>
> A: The selected datasets are representative of MPR benchmarks and are frequently employed by other works, such as Matbench [4]. We have uploaded a revision, where the distribution of atom numbers and lattices and more characteristics for each dataset are provided in Appendix A.1.
>
> > Q2: are there any insights into adapting Noise Contrastive Estimation (NCE) instead of other methods?
>
> A: The proposed matching optimizations focus on modeling the relationship between x and y within the training set, this allows the model to learn the compatibility between x and y, enabling it to infer the compatibility between unseen x and unseen y. We chose to adapt NCE due to its simplicity and its ability to achieve these goals effectively. Nevertheless, we believe that other methods that can fulfill these objectives are also applicable.
>
> **Reference**
>
> [1] Xu, Keyulu, et al. "How neural networks extrapolate: From feedforward to graph neural networks."
>
> [2] Florence, Pete, et al. "Implicit behavioral cloning."
>
> [3] Shen, Xinwei, and Nicolai Meinshausen. "Engression: Extrapolation for nonlinear regression?."
>
> [4] Dunn, Alexander, et al. "Benchmarking materials property prediction methods: the Matbench test set and Automatminer reference algorithm."

---

> > ### Comment · Reviewer_JB71 · 2024-11-27
> >
> > Thanks for your response. I have read the rebuttal carefully. I still have questions as follows:
> > - The comparison of the running time between between LDS and MEX is not fair. MEX requires the other properties of the material. While to get these properties can cost a lot of time.
> > - For the question "more DL-based methods should be compared". Although extrapolation remains an underexplored area, structure-based material property prediction methods are well explored, like ALIGNN, CGCNN, and so on. Besides, structure-based methods only require the crystal structure while MEX requires additional property labels.
> >
> > Thanks for your response, due to the reasons above, I will remain my evaluation.

---

> > > ### Author Response · Authors · 2024-11-27
> > >
> > > Thank you for your comments. We hope the following responses address your misunderstandings.
> > >
> > > > Q: The comparison of the running time between between LDS and MEX is not fair.
> > >
> > > A: The comparison is completely fair as the running time shown in the table represents the total time required by MEX, **encompassing the entire inference process**. During inference, MEX only requires (1) the material structure and (2) $C$ single-value scalars uniformly sampled from a pre-defined range $[a,b]$, e.g., $[−10,10]$. This procedure involves no additional time-consuming steps, as the scalar values are generated directly and do not depend on computing other material properties.
> > >
> > > > Q: Although extrapolation remains an underexplored area, structure-based material property prediction methods are well explored, like ALIGNN, CGCNN, and so on.
> > >
> > > A: The examples provided by the reviewer, ALIGNN and CGCNN, are backbone architectures for material structure encoders **rather than extrapolation methods**. We emphasize that MEX is a general framework that is agnostic to the choice of backbone. In our work, we selected two representative and high-performing backbones—PaiNN and EquiformerV2—that are sufficient to validate the effectiveness of MEX in the context of our experiments.
> > >
> > > > Q: Besides, structure-based methods only require the crystal structure while MEX requires additional property labels.
> > >
> > > A: The additional input labels serve as the foundation for MEX and facilitates extrapolation. They are randomly sampled from a pre-defined interval and the process is computationally efficient.

---

> > > > ### Comment · Reviewer_JB71 · 2024-11-27
> > > >
> > > > Thanks for the authors' response. I understand the property value is randomly sampled.
> > > >
> > > > - As mentioned in the paper "We define MPR extrapolation tasks as predicting unobserved material property values that lie outside the training label range.", all other structure-based DL methods can be applied to this problem. What the authors need to do is splitting the dataset into training, evaluating, and testing following the settings of this task.
> > > >
> > > > - Given that the initial property value is uniformly sampled from a pre-defined range, the accuracy of prediction is limited by the sampling density. Besides, in contrastive learning, the OOD property values are taken as negative samples, how can they be identified as the property for one of the materials?
> > > >
> > > > - This framework requires a pre-defined range. How can we predefine a range for real problems?

---

> > > > > ### Author Response · Authors · 2024-11-27
> > > > >
> > > > > Thanks for your comments. We have addressed your questions in the following response.
> > > > >
> > > > > > Q1: What the authors need to do is splitting the dataset into training, evaluating, and testing following the settings of this task.
> > > > >
> > > > > A: Our disjoint splitting indeed follows the settings of the extrapolation task. Specifically, a disjoint split between validation/testing and training ensures that the task involves "predicting unobserved material property values that lie outside the training label range".
> > > > >
> > > > > > Q2.1: the accuracy of prediction is limited by the sampling density.
> > > > >
> > > > > A: We acknowledge that the prediction precision is influenced by the number of candidate labels $C$. However, a precision-efficiency balance can be achieved by selecting an appropriate value for $C$. This trade-off has been discussed in detail in the "scalability and computational complexity" part of our response to the weaknesses you raised. As demonstrated there, MEX could maintain strong performance while ensuring computational efficiency.
> > > > >
> > > > > > Q2.2: the OOD property values are taken as negative samples, how can they be identified as the property for one of the materials?
> > > > >
> > > > > A: Note that negative samples are not necessarily OOD property values. As we discussed in Section 3.2.1, we select values near the golden value as negative samples.
> > > > >
> > > > > > Q3: This framework requires a pre-defined range. How can we predefine a range for real problems?
> > > > >
> > > > > A: We can predefine such a range for realistic MPR tasks because material properties are governed by quantum mechanics, and their feasible value range is inherently restricted by physical laws. This allows us to reasonably estimate the bounds. A similar question was raised by Reviewer N2Xe, and we also have tested our method with broader bounds in Q3.2.

---

> > > > > > ### Comment · Reviewer_JB71 · 2024-11-27
> > > > > >
> > > > > > Thanks for your response. I do not think the authors' responses addressed my concern.
> > > > > >
> > > > > > For Q1: "all other structure-based DL methods can be applied to this problem. What the authors need to do is splitting the dataset into training, evaluating, and testing following the settings of this task.". So the authors can also apply other models like CGCNN, and ALIGNN directly to this problem without their MEX.
> > > > > >
> > > > > > For Q2.1: The success rate is still poor.
> > > > > >
> > > > > > For Q2.2:  The authors mentioned in Section 3.2.1 that they use some mixing Gaussian techniques to sample negative samples randomly. However, the values can never be a positive sample since this is an extrapolation task, where the OOD values should never exit in the training.
> > > > > >
> > > > > > For Q3: Are all the predefined ranges used in this paper defined according to quantum mechanics? If not, how does the range affect the experimental results?

---

> ### Author Response · Authors · 2024-11-22
>
> Dear Reviewer JB71,
>
> Thank you once again for your valuable and constructive feedback on our submission. As the discussion period progresses toward its end, we kindly request any additional comments or thoughts you may have regarding the clarifications and results we have provided in our rebuttal. We fully understand that this is a busy time, and we sincerely appreciate the effort and time you have dedicated to reviewing our work. If there are any remaining questions or concerns, we would be happy to address them promptly.
>
> Best regards, The Authors

---

> ### Author Response · Authors · 2024-11-25
> **Look forward to your post-rebuttal feedback!**
>
> Dear Reviewer JB71,
>
> Thanks again for your insightful suggestions and comments. Since the deadline of discussion is approaching, we are happy to provide any additional clarification that you may need. In our previous response, we have carefully studied your comments and made detailed responses summarized below:
> - Provide further explanation of the novelty and rationality of our method.
> - Perform additional experiments to demonstrate the computational complexity of our method.
> - Explain the choice of baseline methods and datasets.
>
> We hope that the explanation of our method and the provided new experiments have convinced you of the contributions of our submission.
>
> Please do not hesitate to contact us if there's additional clarification or experiments we can offer. Thanks!
>
> Thank you for your time!
>
> Best, Authors

---

### Official Review · Reviewer_oJTR · 2024-11-11

**Soundness:** 2
**Presentation:** 2
**Contribution:** 2
**Rating:** 5
**Confidence:** 3

**Summary:**

(Note: The reviewer does not have a background in material property prediction; thus, this review is based on informed estimates. The reviewer welcomes discussions and is open to adjusting scores or comments based on feedback from the authors and other reviewers.)

The paper introduces a novel approach that reframes material-property regression (MPR) as a material-property matching problem, aiming to simplify target function complexity. This reframing addresses the difficulty neural networks face in capturing complex non-linearity beyond the training data, improving model extrapolation.

The core idea is that focusing on the proximity between material and property representations, rather than on precise value predictions, reduces learning difficulty and enhances extrapolation. The authors propose two objectives for learning aligned feature spaces for material-property representation matching. First, they use absolute matching optimization with a negative cosine similarity loss to pull paired material and label representations closer together. Second, the method employs Noise Contrastive Estimation (NCE) to help the model distinguish target from noisy labels, thereby capturing fine-grained relative matching relationships.

Within these well-aligned latent spaces, the proposed method (MEX) predicts by optimizing for the nearest target value for a given sample. Experiments demonstrate that MEX not only performs best on the benchmark but also shows strong detection capabilities for promising materials, underscoring its extrapolation potential and suitability for robust material discovery.

------
While reading this paper, I hypothesized that the label encoder could easily overfit to the training data, effectively reducing it to a look-up table and thereby losing any extrapolation capability. However, this risk may be mitigated by the inclusion of a noisy label component, which introduces stochasticity to the model, reducing its tendency to overfit. And the Gaussian applied to the label likely encourages continuity in the label space, which could help to foster the model’s extrapolation abilities.

**Strengths:**

•	The approach addresses a significant problem in material property prediction.

•	The results look promising

**Weaknesses:**

•	Dataset Limitations: The dataset is small and simplistic, limiting the evaluation of the method's effectiveness. The number of samples (ranging from 4,764 to 18,982) is limited, and details about the dimensionality of data points are not provided. Additionally, the design of y target $y_{\text{target}}$, as described in section 3.1, seems unrealistic since the training and target data are entirely disjoint. This choice could disadvantage baseline methods.

•	Baseline Choice: The Deep Imbalanced Regression (DIR) technique is designed for handling imbalanced data distributions with underrepresented target values. However, the proposed dataset’s disjoint target-training setup may hinder DIR’s performance. DIR methods are not specifically tailored for extrapolation, which is central to this work, making it challenging to evaluate against MEX.

**Questions:**

1.	Could the authors clarify the Geometric Mean metric to aid readers unfamiliar with it?
2.	In Figure 6, the performance seems to decrease as $\lambda$ increases, suggesting that NCE might negatively impact results. Could the authors explain this trend?
3.	The authors claim that the matching-based approach enhances extrapolation, but how does this method handle out-of-distribution data or outliers? Can contrastive learning effectively handle these cases?
4.	Could the authors consider an additional setting where the target and training data overlap slightly, such as by adding extreme high or low values to the training set, to simulate an imbalanced distribution?
5.	How well does the method scale with higher-dimensional targets?
6.	Is there a threshold in the matching function M(x, y) to determine when a match is strong enough?
7.	How does the model avoid overfitting in the label encoder, potentially transforming it into a look-up table? Does the noise component mitigate this risk, supporting extrapolation?
8.	While $y^*$ is estimated via Monte Carlo sampling, could gradient-based methods be viable for this estimation?
9.	What is the dimensionality of the input data sample x?

---

> ### Author Response · Authors · 2024-11-19
> **To Reviewer oJTR (Part Ⅰ)**
>
> We thank the reviewer for the detailed and thoughtful review comments. We have addressed the concerns raised  and humbly hope that the reviewer will consider increasing the evaluation score.
>
> **Response to weaknesses**
> > 1. Dataset Limitations
>
> **Dataset choice**: While our dataset size may seem small compared to large-scale CV and NLP datasets, it is standard for MPR benchmarks. As we stated in Section 4.1, we curate our datasets from Matminer,  one of the most widely used tools for materials data, with over 750+ citations.
>
> **Dimensionality of data points**: In MPR, material samples are not represented as continuous vectors like image or word embeddings. Instead, a sample material with $n$ atoms is represented by $(h_i,x_i)_{i=1}^n$, where $h_i\in\mathbb{N}^{+}$ denotes the atom type and $x_i\in\mathbb{R}^3$ represents the Cartesian coordinates of the $i$-th atom. The materials in our dataset are crystal, so they have additional information such as the lattice matrix ($3\times 3$ matrix capturing the crystal's fundamental translation symmetry). This raw information will be transformed into a graph for further processing. We have uploaded a revision, where more characteristics for each dataset are provided in Appendix A.1, to further clarify the details of our dataset.
>
> **Design of $y_{target}$**:  The decision to separate training and target data fully aligns with our objective of evaluating true extrapolation capabilities. Extrapolation remains a significant challenge for neural networks, and prior studies [1,2,3] have explored this issue with the same disjoint data configurations, though typically for simpler data structures. In materials science, extrapolation is particularly crucial, as researchers aim to discover materials with properties beyond those observed to date, driving advancements in technology and innovation.
>
> > 2. Baseline Choice
>
> While DIR methods are not specifically designed for extrapolation, our baseline LDS and Ranksim claim their effectiveness in this area and report performance on disjoint label ranges. To date, as extrapolation is an underexplored challenge in deep learning, we believe that DIR remains one of the most viable approaches for addressing extrapolation in the absence of purpose-built methods. If the reviewer has suggestions for additional suitable baselines, we will try our best to provide experimental results.

---

> ### Author Response · Authors · 2024-11-19
> **To Reviewer oJTR (Part Ⅱ)**
>
> **Response to questions**
>
> > Q1: clarify the Geometric Mean metric
>
> A: This metric was first introduced in [4] and is defined as $(\Pi_{i=1}^N e_i)^{1/N}$, where $e_i:=|y_i - \hat{y}_i|$ represents the L1 error of each sample. As described in [4], "GM aims to characterize the fairness (uniformity) of model predictions using the geometric mean instead of the arithmetic mean over the prediction errors." This metric is widely used in DIR research.
>
> > Q2: the performance seems to decrease as $\lambda$ increases, suggesting that NCE might negatively impact results
>
> A: We would like to first clarify that $\lambda$ represents the weight of the absolute matching relationship $L_{abs}$ rather than the relative matching relationship $L_{NCE}$. As $\lambda$ increases, $L_{abs}$ may dominate the optimization process, causing the model to struggle in distinguishing between negative and positive values, which can affect performance.
>
> > Q3: how does this method handle out-of-distribution data or outliers?
>
> A: Our matching-based approach focuses on aligning the sample and label feature spaces, making it less dependent on the sample distribution. Even when presented with an outlier sample, our method can still effectively optimize the model by aligning the features of the sample with those of its corresponding label.
>
> > Q4: Could the authors consider an additional setting where the target and training data overlap slightly?
>
> A: We follow the reviewer's suggestion and consider a setting where training labels overlap slightly with the target. Specifically, we randomly chose 10 samples from the validation and test set separately and moved them to the training set, forming a DIR-like scenario. We compared our method and SOTA baseline methods in the following table. We can observe that (1) both methods exhibit a performance improvement compared to the original setting and (2) our method is still superior to SOTA baselines in most tasks.
>
> |Metric     | Algorithm  | Formation Energy (bottom) | Shear Modulus (bottom) | Shear Modulus (top) | Refractive Index (bottom) | Refractive Index (top) |Phonons Mode Peak (bottom) |Phonons Mode Peak (top)
> |:--|:--|:--|:--|:--|:--|:--|:--|:--|
> |MAE($\downarrow$)|SOTA Baseline|0.228(0.008) |0.364(0.017) |**0.181(0.021)** |0.190(0.012) |0.553(0.01) |0.417(0.012) |**0.331(0.035)** |
> |                 |MEX          |**0.162(0.01)** |**0.356(0.003)** |0.236(0.012) |**0.136(0.006)** |**0.531(0.008)** |**0.414(0.023)** |0.362(0.021)|
> |GM($\downarrow$)|SOTA Baseline |0.179(0.009) |0.277(0.023) |**0.125(0.026)** |0.105(0.01) |0.406(0.016) |**0.283(0.012)** |0.22(0.036)  |
> |                 |MEX          |**0.106(0.008)** |**0.265(0.006)** |0.183(0.01) |**0.062(0.003)** |**0.388(0.027)** |0.306(0.035) |**0.219(0.02)**  |
>
> > Q5: How well does the method scale with higher-dimensional targets?
>
> A: Theoretically, our method can be extended to higher-dimensional targets with minimal modifications for negative target sampling. A straightforward approach would be to scale the one-dimensional Gaussian in Eq. 4 to a multi-dimensional Gaussian. However, since material target properties are typically single-dimensional, we leave the exploration of this extension in future work.
>
> > Q6: Is there a threshold in the matching function M(x, y) to determine when a match is strong enough?
>
> A: While MEX is inherently threshold-free, users can define a threshold using heuristic rules, such as setting it to the average M(x, y) value from the training set samples.
>
> > Q7: How does the model avoid overfitting in the label encoder, potentially transforming it into a look-up table? Does the noise component mitigate this risk, supporting extrapolation?
>
> A: We sincerely thank the suggestion. However, it is non-trivial to implement a look-up table in this context, as such approaches are generally suited for discrete labels, whereas our task involves continuous material properties. The noise component indeed plays a critical role in reinforcing proper matching relationships between labels and an anchored sample, which implicitly maintains inter-label consistency, thereby mitigating the risk of overfitting and supporting robust extrapolation.
>
> > Q8: could gradient-based methods be viable for this estimation?
>
> A: Certainly, gradient ascent and other optimization methods could be applied for inference in our framework. We opted for Monte Carlo sampling due to its derivative-free nature, which allows for efficient inference while maintaining strong performance.
>
> > Q9: What is the dimensionality of the input data sample x?
>
> A:  This question has been addressed in weakness 1.
>
> **Reference**
>
> [1] Xu, Keyulu, et al. "How neural networks extrapolate: From feedforward to graph neural networks."
>
> [2] Florence, Pete, et al. "Implicit behavioral cloning."
>
> [3] Shen, Xinwei, and Nicolai Meinshausen. "Engression: Extrapolation for nonlinear regression?."
>
> [4] Yang, Yuzhe, et al. "Delving into Deep Imbalanced Regression."

---

> > ### Comment · Reviewer_oJTR · 2024-11-23
> > **Further feedback**
> >
> > Thank the authors for providing their responses to my concerns about their work. Some of them have been addressed, which I encourage the authors to incorporate into the paper, such as:
> >
> > - The dimensionality of the data points
> > - The design of \( y_{\text{target}} \), the choice of baselines (deep imbalanced regression techniques), and the limitations of the baselines in this specific extrapolation setting
> > - The metrics used in this setting
> >
> > Regarding the novelty of the work, again, because I do not have experience or expertise in this field, I will let other reviewers and the area chair decide on the novelty of the method. In my opinion, if extrapolation prediction in material science is important, the contribution should be weighted more toward introducing a new dataset for that problem rather than proposing a novel method. With this mindset (again, from my narrow viewpoint), the dataset in this work is somewhat limited because it is curated from another dataset designed specifically for a different problem.
> >
> > There are also a few concerns that I still do not fully understand. Specifically:
> >
> > - **The contribution of \( L_{\text{abs}}:**
> >   As shown in Figure 6, when \( \lambda \) increases, the performance decreases. It seems that \( L_{\text{abs}} \) contributes negatively to the overall performance, raising the question of whether this component is necessary. From my understanding of self-supervised learning—and contrastive learning in particular—the relative loss is sufficient to construct a meaningful latent space (inducing locality or globality properties).
> >
> > - **Q4:**
> >   Thanks to the authors for conducting additional experiments. What is the SOTA baseline method in that table?
> >
> > - **Q7:**
> >   As evidenced by the negative impact of the \( L_{\text{abs}} \) loss, I feel that the main contribution to the success of the proposed method is \( L_{\text{NCE}} \), particularly the introduction of label noise sampled from a Gaussian distribution. From Figure 3 and Table 2, I observed that the Formation Energy and Refractive Index settings are quite similar to a Gaussian distribution and, incidentally, are also the settings in which the proposed method achieves good performance. Can the authors provide further discussion on this observation?

---

> > > ### Author Response · Authors · 2024-11-25
> > > **To Reviewer oJTR (Part Ⅲ)**
> > >
> > > Thank you for your insightful feedback. We have incorporated your suggestions into the revised manuscript and provided detailed explanations for each question below.
> > > > Q1:  the dataset in this work is somewhat limited because it is curated from another dataset designed specifically for a different problem.
> > >
> > > We understand the reviewer’s concern and would like to emphasize the following: (1) The original dataset consists of material-property samples without a predefined split configuration, offering flexibility to adapt it for addressing specific research objectives. (2) Redefining dataset splits to align with new research settings is a widely adopted practice. For example, in DIR tasks, authors often reconfigure datasets to exhibit significant data imbalance. Similarly, for tasks such as few-shot learning, zero-shot learning, and out-of-distribution (OOD) detection, redefining splits is a standard approach to create scenarios that reflect the intended problem setting. Therefore, we believe that resplitting is appropriate for effectively exploring the extrapolation challenges in material science.
> > >
> > > > Q2: about the contribution of $L_{abs}$
> > >
> > > To address the reviewer's concern about $L_{abs}$, we expand its weight range from [0.25, 1] to a larger range [0.01, 1] to observe its impact on performance.  As shown in Figure 6 of the revised paper, with the increase of the weight, performance increases first and then decreases for most settings, highlighting the necessity of $L_{abs}$ in achieving optimal results. Additionally, we report the performance when $L_{abs}$ is set to 0 (i.e., using only NCE). In this case, during inference, we replace the cosine similarity with the NCE score function to estimate sample-label matching as the cosine relation is not optimized without $L_{abs}$ . As shown in the table below, this modification significantly limits performance, further demonstrating the importance of $L_{abs}$ in the overall framework.
> > >
> > > | |Formation Energy (bottom) | Shear Modulus (bottom) | Shear Modulus (top) | Refractive Index (bottom) | Refractive Index (top) | Phonons Mode Peak (bottom) | Phonons Mode Peak (top) |
> > > |--|--|--|--|--|--|--|--|
> > > |w $L_{abs}$| 0.172(0.008) |0.376(0.010) | 0.245 (0.020) | 0.141(0.004) |0.501(0.018) |0.495(0.007)  |0.789(0.011) |
> > > |w/o $L_{abs}$ | 0.47(0.013) | 0.525(0) | 0.323(0.001) | 0.244(0.012) | 0.611(0.01) | 0.858(0.046)| 0.917 (0.011)|
> > >
> > > > Q3: What is the SOTA baseline method in that table?
> > >
> > > We compared our method with SOTA baselines based on their primary results presented in Table 2. Specifically, we selected LDS for Formation Energy, ConR for Phonons Mode Peak (top), and BalancedMSE for the remaining datasets.
> > >
> > > > Q4: the Formation Energy and Refractive Index settings are quite similar to a Gaussian distribution and, incidentally, are also the settings in which the proposed method achieves good performance.
> > >
> > > $L_{abs}$ is necessary as we demonstrated in Q2. Additionally, we appreciate the reviewer’s insightful observation regarding the similarity between the noise distribution and the dataset distribution. To investigate the impact of this similarity on performance, we replaced the Gaussian noise distribution with a uniform distribution over the label search interval. As shown in the table (MAE) below, the modified MEX achieves performance comparable to the original setting (even better), indicating that the similarity is not the underlying reason for the success of our method.
> > > |Noise distribution | Formation Energy (bottom) | Refractive Index (bottom) | Refractive Index (top) |
> > > |--|--|--|--|
> > > |Gaussion | 0.172(0.008)  | 0.141(0.004)  | 0.501(0.018) |
> > > |Uniform | 0.154(0.019) | 0.145(0.008) | 0.441(0.025) |

---

> > > > ### Comment · Reviewer_oJTR · 2024-11-27
> > > > **Further feedback**
> > > >
> > > > Dear Authors,
> > > >
> > > > Thank you for providing additional experiments to address my concerns regarding L_abs (Q2) and the distribution matching (Q4). I appreciate your efforts to clarify these points.
> > > >
> > > > While the proposed method appears technically capable of addressing the extrapolation problem, my overall impression of this work is that the self-curated extrapolation dataset is somewhat limited. I understand that proposing a new dataset or benchmark is challenging and requires significant resources and effort. The mindset may differ in material science or other specialized fields, but in this field (in my opinion)—and particularly at this conference—there is often greater appreciation for either a simple method that is effective on real-world datasets or a technically novel method that just works on simple synthetic datasets. Unfortunately, the approach proposed in this paper, while simple and effective, is demonstrated only on simple synthetic datasets.
> > > >
> > > > My humble/naive suggestion to improve the paper is to increase the complexity of the datasets. If there is no prior work, the paper could be a benchmark to the field, and thus be a significant contribution.
> > > >
> > > > With the above concern, I would like to keep my original evaluation.
> > > >
> > > > Thanks

---

> ### Author Response · Authors · 2024-11-22
>
> Dear Reviewer oJTR,
>
> Thank you once again for your valuable and constructive feedback on our submission. As the discussion period progresses toward its end, we kindly request any additional comments or thoughts you may have regarding the clarifications and results we have provided in our rebuttal. We fully understand that this is a busy time, and we sincerely appreciate the effort and time you have dedicated to reviewing our work. If there are any remaining questions or concerns, we would be happy to address them promptly.
>
> Best regards, The Authors

---

### Meta-Review · Area_Chair_Xpei · 2024-12-21

**Metareview:**

This work introduces a method named Matching-based Extrapolation for material property regression (MPR) using deep learning. The authors highlight the shortcomings of traditional MPR approaches, particularly the reliance on the i.i.d. assumption, and argue that extrapolation is crucial for MPR. To address this, they propose a method employing two encoders: a material property encoder and a label encoder, which are learned through alignment. Additionally, they introduce a score-matching module that evaluates the compatibility between a material and a given label, providing an extra supervisory signal for the training of the encoders. At inference time, the similarity between the encoded features of a material and the label encoder is maximized via Monte Carlo sampling, enabling predictions beyond the range of training labels.

The authors evaluated their method against previous approaches, curated a dataset to assess extrapolation capabilities, and provided performance comparisons. However, reviewers raised several concerns, including the limited scale of the experiments, the experimental setup (e.g., knowledge about the range of labels), and benchmarking practices. While the authors responded to these concerns during the discussion period, the reviewers were not fully convinced, and none championed the paper.

The AC concurs with the reviewers that a re-design of the experiments might be necessary to strengthen the work. Consequently, and sadly, the AC recommends `rejection.`

**Additional Comments On Reviewer Discussion:**

The paper was reviewed by three reviewers, who raised concerns about the experiments, computational complexity of the algorithm, and benchmarking. While some of these concerns were addressed by the authors during the author-reviewer discussion period, others remained unresolved.

Following the discussion, the reviewers and the AC further deliberated on the paper. None of the reviewers recommended acceptance, as several issues remained unresolved in its current form. The AC agrees with the reviewers that, despite its merits, the work is not yet ready for publication. Therefore, the AC recommends rejection.

---

### Decision · Program_Chairs · 2025-01-22

Reject